# Some Reflections on Drawing Causal Inference using Textual Data: Parallels Between Human Subjects and Organized Texts

**Bo Zhang**                                          BOZHAN@WHARTON.UPENN.EDU
*Department of Statistics and Data Science*
*University of Pennsylvania*
*Philadelphia, PA 19104, USA*

**Jiayao Zhang**                                      ZJIAYAO@WHARTON.UPENN.EDU
*Cognitive Computation Group*  and
*Department of Statistics and Data Science*
*University of Pennsylvania*
*Philadelphia, PA 19104, USA*

**Editors:** Bernhard Schölkopf, Caroline Uhler and Kun Zhang

## Abstract

We examine the role of textual data as *study units* when conducting causal inference by drawing parallels between human subjects and organized texts. We elaborate on key causal concepts and principles, and expose some ambiguity and sometimes fallacies. To facilitate better framing a causal query, we discuss two strategies: (i) shifting from immutable traits to perceptions of them, and (ii) shifting from some abstract concept/property to its constituent parts, i.e., adopting a constructivist perspective of an abstract concept. We hope this article would raise the awareness of the importance of articulating and clarifying fundamental concepts before delving into developing methodologies when drawing causal inference using textual data.

**Keywords:** Causal inference; Constructivism; Natural language processing; Potential outcomes framework; Pretreatment variables

## 1. Introduction: Causal Inference with Textual Data

With the unprecedented empirical success of transformers in various natural language processing tasks spanning from text comprehension to machine translation, extracting and representing semantic meanings from textual data have witnessed the rise to a new greatness (Devlin et al., 2019; Liu et al., 2019; Radford et al., 2019). Recently, there is an abundance of interest in drawing causal conclusions using high-dimensional, structured or less structured, contextual or less contextual, representations of textual data in one way or another. There are at least three[1] roles of textual data in a causal query (Feder et al., 2021; Sap et al., 2020).

(i) Textual data serve as vehicles for reasoning (commonsense) causal relations of semantic meanings (Ning et al., 2018; Sap et al., 2020; Shwartz et al., 2020; Zhang et al., 2022).

(ii) Textual data encode important covariates or outcome of interest in human populations research (Egami et al., 2018). For instance, in a study of the treatment effect of a new technology in cardiac surgery, patients' medical history hand-written by their physicians could encode patients' important clinical characteristics, which are likely to confound medical decisions and clinical outcomes and should be adjusted for.

---

1. For example, there is also much interest in *causal model explanation/interpretation* that aims to understand a notion of model sensitivity, which is different from causal inference in the context of this essay.

(iii) Textual data are themselves *study units* in a causal query, and interests lie in intervening in some aspect of textual data (Veitch et al., 2020; Feder et al., 2021). For instance, Veitch et al. (2020) studied the "causal effects" of number of theorems in a conference paper on its acceptance rate; Sridhar and Getoor (2019) investigated if the tone of replies results in a change of sentiment using transcripts from online debates; Roberts et al. (2020) studied if authors' gender affects articles' citation in a large cohort of academic articles on international relations. Keith et al. (2020, Table 1) surveyed many additional causal queries involving textual data as study units.

The primary goal of this article is to discuss the role of text data as *study units* (role (iii) discussed above) in causal inference by drawing some useful parallels between organized textual data and human subjects understood broadly, the most typical study units in classical causal inference literature; see, e.g., Cochran and Chambers (1965), Rosenbaum (2002b), and Imbens and Rubin (2015). In particular, we (i) examine key concepts, including causal variables and covariates, associated with textual data, (ii) discuss some ambiguity, inconsistency, and common fallacies we have observed in the machine learning (ML) and natural language processing (NLP) literature, and (iii) summarize two useful strategies to help researchers better frame their causal queries when study units are textual data.

It perhaps comes as no surprise that much ambiguity we identified in the ML/NLP literature when study units are textual data was also encountered in human populations research in classical causal inference literature; hence, we found it valuable to bridge the classical causal inference literature, including works in statistics, biostatistics, economics, political science, etc, and the ML/NLP literature, so that some time-honored wisdom from pioneers in causal inference would benefit researchers in the ML/NLP field, and novel ideas and recent progress from ML/NLP and the abundance and ubiquity of textual data could also benefit causal inference practitioners in biomedical and social sciences. We will ground our discussion in the potential outcomes framework (Neyman, 1923; Rubin, 1974; Imbens, 2020), though many parallels between textual data and human subjects and caveats we derived should also be useful and readily available to other causal paradigms (Pearl, 1995, 2009; Heckman, 2005; Peters et al., 2017). We will use two examples discussed in Veitch et al. (2020) to illustrate basic concepts, principles, and sometimes fallacies. We have nothing against the authors; on the contrary, we believe that these authors, in Veitch et al. (2020) and many related works, have made many pioneering and useful contributions to expanding the boundary of causal inference to textual data.

## 2. Study Units and Covariates

One caveat Donald Rubin gave to practitioners of causal inference (and statistics at large) is that researchers should articulate the quantity of interest, i.e., *estimands*, prior to describing an algorithm and its associated output (Rubin, 2005).

In causal inference problems, a unit-level causal effect refers to a contrast in potential outcomes of a study unit, and a summary causal effect, e.g., the sample average causal effect, refers to causal effects averaged over a collection of units. The first key concept involved in defining a causal effect is the notion of "study units." For instance, in an analysis of the effect of utilizing a certain technology in cardiac surgery using the U.S. Medicare and Medicaid data, units involved in the study are mostly senior Americans aged over 65 (MacKay et al., 2021; Zhang et al., 2021a). A causal query where each study unit is clearly a complete piece of article, including all words, figures,

tables, references, and metadata like authors, their affiliations, etc, can be found in the following example in Veitch et al. (2020).

**Example 1** Consider a corpus of scientific papers submitted to a conference. Some have theorems; others do not. The goal is to infer the causal effect of including a theorem on paper acceptance. The effect is confounded by the subject of the paper: more technical topics demand theorems, but may have different rates of acceptance. The data do not explicitly list the subject, but it does include each paper's abstract. It is hoped to use the text to adjust for the subject and estimate the causal effect.

Let us recall that associated with each study unit is a vector of, possibly high-dimensional, "covariates." According to Rubin (2005), covariates refer to

> *"variables that take their values **before** the treatment assignment or, more generally, simply **cannot be affected** by the treatment, such as preaspirin headache pain or sex of the unit."*

Not all variables can be regarded as "covariates" in the above sense. A variable is a "covariate" when the study unit takes on the value *before* the treatment assignment. For instance, in the Medicare example, senior Americans' covariates include their race/ethnicity, age, preexisting comorbidities, etc, and these variables qualify as "covariates" because their values cannot be modified by the new technology or the absence of it in the cardiac surgery. To stress the "temporal" nature of "covariates," researchers in statistics, epidemiology, and social sciences often refer to them as "pretreatment variables." It is widely appreciated that "pretreatment variables" need to be adjusted for (Rosenbaum, 2002a), via matching, subclassification, or model-based methods, in order to obtain an unbiased causal estimate of a meaningful causal estimand; on the contrary, concomitant variables affected by the treatment, or the so-called "posttreatment variables" should not be adjusted for. In fact, adjusting for "posttreatment variables" often leads to "posttreatment bias" (Rosenbaum, 1984).

A central question then emerges when we start drawing parallels between two types of study units: human subjects and organized textual data:

> *What qualify as "covariates" or "pretreatment variables" when the study units involved in the analysis are textual data organized in one way or another?*

## 3. Immutable Characteristics and Causal Variables

To understand what variables quality as "pretreatment variables," we first need to articulate the "treatment" or more generally the "causal variable" under consideration. We would like to argue that many "attributes" of textual data, e.g., number of theorems in a conference paper in Example 1, are *not* appropriate causal variables *per se*.

To continue our analogy between human subjects and textual data, consider the role of "race", "ethnicity", or "gender identity" in human populations research. In a well-argued article, Holland (2008) pointed out that attributes of study units are not "causal variables" if they are immutable and do not lend themselves to "plausible states of counterfactuality." Put a different way, "it is critical that each unit be potentially exposable to any of the causes," and that it is not appropriate to talk about "causation without manipulation" (Holland, 1986). Moreover, if we imagine race/ethnicity, understood biologically as opposed to a social construct, as being "assigned" at a human subject's

conception, then almost every aspect of the human subject, including socioeconomic status, preexisting comorbid conditions, etc, is a "posttreatment variable" with gender being possibly the only exception (King, 1991; Gelman and Hill, 2006).

Organized textual data are different from human subjects in many ways; however, similar concerns persist. Take as an example the number of theorems in a conference paper. In their original article, Veitch et al. (2020) regarded "the sequence of words" in a conference paper as "covariates." From our discussion of race/ethnicity in human populations research, it becomes obvious that this notion of covariates is, at best, untenable and demands much clarification: many words in a conference paper are "affected" by "whether there is a theorem" or the "number of theorems," e.g., those discussing implications of the theorems in the abstract and the conclusion section. The same concern persists and clarification is much needed when embeddings from language models of "the sequence of words" are used as "covariates." Moreover, it is very difficult to even conceptualize exposing each study unit (i.e., a conference paper) to this "treatment:" it is nearly nonsensical to ask what would have happened to a pure empirical/dataset/benchmark paper in, say, EMNLP or ICLR, had it proved one or more theorems. Even when we restrict our attention to methodological papers backed up by simulations but not theorems, it is not clear what theorems these methodological papers would have proved in the first place. In fact, there are likely many different *versions* of this "treatment" of "adding one theorem," e.g., one version being "proving one theorem on the convergence rate of the proposed algorithm" and another version being "proving one theorem on the minimax lower bound of the problem," thus violating the *stable unit treatment value assumption* (SUTVA) as discussed in more detail in Rubin (1980, 1986). The key here is to think twice whether the "causal variable" under consideration lends itself to "plausible states of counterfactuality" (Holland, 2008). Even in some scenarios where adding or subtracting a theorem of a particular type can be envisioned, researchers need to articulate these scenarios and properly restrict their attention to meaningful study units.

Again, borrowing wisdom from classical causal inference reasoning, one acceptable, albeit not entirely satisfying, solution is to adjust for attributes that are definitely not affected by the number of theorems and drop any aspect of the article that are likely to be affected by it. In this spirit, researchers could adjust for certain "metadata" of the conference paper, e.g., topic listed, authors, and their affiliations, but *not* attributes of the article like the number of references, number of equations, sentiment and flow of the article, etc: these aspects are all likely to be affected by the number of theorems. This strategy is sometimes used in human populations research when the causal variable of interest is some immutable trait like "race/ethnicity" or "gender identity" (King, 1991), although better strategies (in our opinion) exist as we are ready to discuss.

## 4. Two Strategies Facilitating Framing Proper Causal Queries

We discuss two strategies that could be useful when researchers would like to properly frame the causal queries when study units are textual data.

### 4.1. Shifting from actual traits to perceptions of them

In a seminal paper, Greiner and Rubin (2011) discussed some prerequisites for the design and analysis of observational studies when the causal variable of interest is immutable. Greiner and Rubin (2011)'s key insight is that there are often *two actors* in causal inquiries concerning immutable traits: a study unit that possesses such immutable traits and a "decider" that *perceives* such traits.

The causal query is often concerned about decider's behaviors after perceiving study units' certain immutable trait. While it is not appropriate to "manipulate" study units' immutable traits, it is possible, both conceptually and in many cases operationally, to manipulate a decider's perception of such immutable traits; similar ideas and arguments were also presented by Fienberg and Haviland (2003) and Kaufman (2008). We illustrate this point using Veitch et al. (2020)'s second example.

**Example 2**  Consider comments from `reddit.com`, an online forum. Each post has a popularity score and the author of the post may (optionally) report their gender. The goal is to examine whether there is a direct effect of a "male" label on the score of the post. However, the author's gender may affect the text of the post, e.g., through tone, style, or topic choices, which also affects its score. Again, it is hoped to use the text (post content) to estimate the causal effect.

In this example, Veitch et al. (2020) imagine "manipulating" the gender identity of the *author* of a post and consider a mediation-type analysis (Imai et al., 2010). This is not a well-posed causal query in our opinion for similar reasons discussed before: it is highly speculative to ask what tone, style, or topic a person would have written in a post at a given time in the life course had the person had a different gender identity.

In fact, Example 2 could be re-formulated under the "perceived traits" framework outlined by Greiner and Rubin (2011). Suppose that some posts are associated with a hashtag or an icon indicating writers' gender identity, then we may stipulate that the treatment happens when the viewer of the post first perceives or reads the post. To stress, the gender identity of the writer of the post is *not* manipulated; viewers' perceptions are manipulated, presumably by manipulating the hashtag or icon attached to the post. By shifting from the actual trait of the study unit, i.e., gender identity of the author of the post, to viewers' perceptions of it, we are able to articulate the timing of the treatment, and it immediately becomes clear what variables qualify as "covariates" for this treatment: any aspect of the post that remains unchanged is a covariate that needs to be adjusted for. This includes pretty much every integral part of a post: words, emojis, pictures, tone, topic, etc. Moreover, it implies what variables do *not* qualify as "covariates:" in short, anything happening after viewers' first perception is a posttreatment and should not be adjustment for, e.g., any aspect of the comments left by the viewers. We note that many authors in the ML/NLP literature have been switching from traits of textual data to perceptions of them; see, e.g., Roberts et al. (2020), Feder et al. (2021), and Pryzant et al. (2021).

### 4.2.  Shifting from one trait to a property/concept including many traits

The question remains as how to frame the causal query in Example 1. Shifting from the number of theorems to viewers' perception of theorems still seems insufficient in this example: it is unclear how to manipulate viewers' perception of theorems without also manipulating viewers' perceptions of other related textual data such as the number of mathematical equations or the complexity of mathematical notation. In this example, presumably researchers are interested in manipulating viewers' perceptions of an article's "mathematical rigor," and "number of theorems" is merely one aspect of this rather general concept.

Let us take a step back and think again the role of "race/ethnicity" in human populations research. A strict biological perspective would regard "race/ethnicity" as being "assigned" at a person's conception; on the contrary, an arguably more appropriate theory of race would hold that

distinction between so-called races are the products of many social forces including cultural, geographical, social, historical, and many other influences (Rutter and Tienda, 2005; Holland, 2008; Sen and Wasow, 2016). The concept of "race/ethnicity," under this "constructivist framework," is a complex consisting of many readily mutable traits, or "a bundle of sticks" (Sen and Wasow, 2016), and causal queries concerning "race/ethnicity" become better-posed from this constructivist perspective of race/ethnicity.

This strategy of further switching from the perception of a concept or one aspect of it to the perception of all constituent parts of the concept could be beneficial to framing causal queries in Example 1 and inferring the causal effect of a certain linguistic property of textual data, e.g., "politeness" as discussed in Pryzant et al., 2021. In Example 1, it seems most appropriate to define the causal variable as viewers' perceptions of an article's mathematical rigor. The notion of mathematical rigor could incorporate many aspects of the article, with the number of theorems being one component. Under this perspective, variables like "number of graphs," "flow of the paper," and "number of references to other theory papers" should be regarded as components of the "causal variable" rather than covariates to be adjusted for. As another example, suppose that we are interested in the causal effect of viewers' perceptions of a linguistic property of textual data, say "politeness." The key issue here is to identify and articulate constituent parts, i.e., "sticks" metaphorically, of the theory of politeness, i.e., the "politeness bundle." For instance, viewers' perceptions of an article/post/email's politeness may consist of their perceptions of its sentiment and formality in wording, among other things. Covariates to be adjusted for may include topic, authorship, brevity, among other things. This task of separating causal variables from covariates should be conducted on a case-by-case basis, and could benefit greatly from linguists' domain knowledge; see, e.g., Brown and Levinson (1978) for a comprehensive theory of politeness.

In any circumstances, we find it important to clearly define the causal variable, including its timing and/or constituent parts when necessary, and articulate what variables qualify as "covariates" for the causal variable under consideration. We recognize that some variables may fall into the grey areas of this "causal variable/covariates" dichotomy; however, acknowledging and articulating the ambiguity is itself a crucial step towards any meaningful scientific communication.

## 5. A Dichotomy of Covariates

Thus far, we have focused mostly on the timing of a treatment and the "causal variables/covariates" dichotomy when study units are textual data. In this section, we assume an appropriate causal query is framed, e.g., how perception of authors' gender identity affects the acceptance of a conference paper, and briefly discuss different types of covariates to be included in such an analysis.

In human populations research, covariates or pretreatment variables are often divided into "observed" and "unobserved" (Rosenbaum, 2002b). When study units are organized textual data, we find it meaningful to further divide observed covariates into two broad categories: "explicit observed covariates" that could be derived from the organized textual data at face value, e.g., the number of theorems/equations/figures in a conference paper, and "implicit observed covariates" that capture deeper aspects intrinsic to the textual data. Some concrete examples of implicit covariates include: bag-of-words embeddings such as Word2Vec (Mikolov et al., 2013) and GloVe (Pennington et al., 2014), and contextual embeddings such as BERT (Devlin et al., 2019) and SentenceBERT (Reimers and Gurevych, 2019); perceived sentiments, tones, and emotions from the text (Barbieri et al., 2020; Pérez et al., 2021); topic modeling and keyword summarizing (Xie et al., 2015; Blei and Lafferty,

2007; Ramage et al., 2009; Wang et al., 2020; Santosh et al., 2020); evaluated trustworthiness of the claims made (Nadeem et al., 2019; Zhang et al., 2021b); temporal relationships and semantic relationships of events mentioned (Zhou et al., 2021; Han et al., 2021); commonsense knowledge reasoning (such as complex relations between events, consequences, and predictions) based on the text (Chaturvedi et al., 2017; Speer et al., 2017; Hwang et al., 2021; Jiang et al., 2021). These are by no means exhaustive; nor are they necessary for each and every causal query. It is always essential to incorporate domain knowledge before determining an appropriate set of covariates.

The central assumption to drawing causal inference from observational data, namely the "treatment ignorability assumption" (Rosenbaum and Rubin, 1983), also known as the "no unmeasured confounding assumption," needs to take into account both explicit and implicit observed covariates, so that the treatment assignment is closer to being randomly assigned within strata defined by both explicit and implicit observed covariates. Moreover, there is an additional layer of complexity as some "implicit observed covariates" (e.g., sentiment of the article) are themselves output from some pretrained language models on certain domains of texts (e.g., twitters) and likely to be measured with error or summarized insufficiently.

Unobserved covariates have a particularly important role in drawing causal inference from non-randomized observational data as Rosenbaum (2018) reminds us that "at the end of the day, scientific arguments about what causes what are almost invariably arguments about some covariate that was not measured or could not be measured." The primary appeal of a randomized controlled experiment is that randomization stochastically balances both observed and unobserved covariates, while conclusions derived from observational data can always be challenged on the basis of unmeasured confounding bias. For instance, Sir Ronald Fisher challenged the causal interpretation of the association between smoking and lung cancer by pointing to the possibility of a genetic variant making a person simultaneously prone to smoking and susceptible to lung cancer (Fisher, 1958). In the context of causal inference with textual data, this concern of unmeasured confounding is even more pronounced as relevant "covariates" like semantics may be buried under the ocean of words and difficult to be fully recovered. Methods that aim to address unmeasured confounding concerns, e.g., sensitivity analysis (Rosenbaum, 2002b, 2010; VanderWeele and Ding, 2017; Veitch and Zaveri, 2020), negative control methods (Lipsitch et al., 2010), and instrumental variable methods (Angrist et al., 1996), should be explored and more actively employed in the context of causal inference with textual data.

## 6. Conclusion

Properly framing a causal query with textual data is challenging. One general caveat that Angrist and Pischke (2008) gave to practitioners of causal inference is that research questions for which there are no experimental analogies (even hypothetical ones in a world with unlimited time, budgets, and omniscient powers) may be fundamentally unidentified or at least ill-posed. We encourage researchers working with textual data to always (i) identify study units, (ii) articulate the treatment or causal variable including its timing or its constituent parts when appropriate, and (iii) make an argument for each covariate to be adjusted for in the analysis. Causal inference is always an ambitious task; carefully going through steps (i)−(iii), though not a panacea, helps at least expose potential fallacies and facilitate conversations and critiques. We hope that our discussion in the article could raise awareness that although methodological development is important, it is equally

important to pay attention to many fundamentals, including key concepts and basic principles, when conducting causal inference with textual data.

## Acknowledgements

This work was supported in part by NSF through CCF-1934876 and ONR Contract N00015-19-1-2620. We would like to thank Dan Roth, Dylan S. Small, and Weijie J. Su for stimulating discussions and helpful feedback on this manuscript.

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
