# OpenReview forum: "Some Reflections on Drawing Causal Inference using Textual Data: Parallels Between Human Subjects and Organized Texts"
_cclear.cc/CLeaR/2022/Conference — CLeaR 2022 Poster_

### Official Review · Reviewer_JrEN · 2021-11-19

**Confidence:** 3
**Overall Score:** 6

**Main Review:**

I would like to start the review by mentioning that this article reads more like a perspective piece than an original research article. This is perfectly fine as far as I am concerned, but I am not sure if the submission then fits this conference's call for papers.

I find that this article is of decent quality and provides a useful perspective on how causal inference methodology can be applied to textual data. The strategies proposed are sound and have value beyond the textual domain. As far as I can tell, the authors manage to include a reasonably comprehensive overview of the current state-of-the-art causal inference methods and applications for textual data. Finally, the paper is very well written and structured, and virtually error-free.

The scope of the article is perhaps a bit limited. The authors focus on causal effect estimation via covariate adjustment, and do not touch upon tasks such as causal structure learning. It might be interesting to include more facets of causal inference into the discussion.

Overall, I think this article is an interesting perspective on the current and future use of causal inference methodology on textual applications, and would be a reasonable contribution to the conference.

Minor comments:
- page 10: "articulate treatment or causal variable" -> "articulate the treatment or causal variable"

**Summary:**

The authors discuss the application of causal inference methods to textual data, and propose two strategies for framing causal questions in a more precise manner: (i) shifting from immutable traits to perceptions of them, and (ii) shifting from abstract concepts to its constituent parts. They discuss how these strategies address the pecularities pertaining to textual data and how they can help in more clearly defining causal inference problems.

---

> ### Author Response · Authors · 2021-12-03
> **Response to Reviewer JrEN**
>
>
> Thanks for reviewing our paper and for your comments. We are glad that you found our paper interesting and appreciate your suggestions and comments. Thanks for pointing out the typo, we have corrected it in the revision. Please find below our per-point responses, and we hope they address your concerns.
>
> ### Goodness-of-fit to CLeaR’s call for papers.
>
> Our motivation for this paper is two-fold. We have observed an increasing trend (especially the NLP community) of exploring the intersection of machine learning models and causal inference. On one hand, we think many of these pioneering attempts would potentially exert very significant impact towards not only the NLP domain, but also to artificial general intelligence; on the other hand, however, we feel that some work tend to use "causality" and "causal inference" too liberally and sometimes without enough justifications for causal assumptions/principles. The goals of our paper are thus
>
> 1. to take several concrete examples and illustrate what causal queries might be less well-formulated, and why we should watch out for them; and
>
> 2. how proper causal queries could be framed while honouring assumptions/existing wisdoms of causal inference.
>
> In this regard, we wish to respond to the call on the “methodology” and “applications” relevant to drawing causal inference using textual data. We hope our paper could be of some help for practitioners to “build machine learning on causal principles” by pointing out potential pitfalls and danger zones, and provide mitigation strategies.
>
> ### Scope of the article.
>
> Thank you for raising this concern as well as pointing out potential topics! We agree with you that in this paper we only used the problem of estimating causal effects as examples while not touching upon other problems such as causal structure learning as you mentioned. The reasons for our choice are mainly as follows,  and we have revised our paper to highlight these and to discuss the applicability of the them to other causal problems.
>
> 1. Estimating causal effect is perhaps one of the most ubiquitous problems in causal inference, contemplated by practitioners of every school of causal formulations, whose problem statement is easy to understand. We think by using it as an example, it will be easier and clearer to illustrate potential pitfalls, and make it more available to a broader range of readers.
>
> 2. Causal effect estimation is the topic that most articles at the intersection of NLP and causality are concerned and familiar with, hence we chose it as our primary focus.
>
> 3. The potential pitfalls and mitigation strategies we discussed were based on the causal model itself, thus relatively less dependent on the specifics of the problem itself: no matter whether one wants to estimate the causal effect or perform causal discovery, regardless of which school of causal models is being used, fundamental questions such as mutable vs. immutable traits, perceptions of treatments, validations of the assumptions (SUTVA, positivity, strong ignorability, etc) need to be addressed. For example, in causal discovery and causal structure learning, by the same argument, researchers are advised to think twice if a certain immutable trait (e.g., race/ethnicity) is found to be a “causal” variable. Another implication of practical importance for causal discovery research is that researchers may need to study how to discover a collection of variables as “causal variables,” e.g., the number of theorems in a paper *plus* the number of equations PLUS reference to theory papers as in our running example.
>
>
> In this sense, we think our discussions can hopefully also benefit practitioners interested in other forms of causal problems such as causal structure learning.

---

> > ### Comment · Reviewer_JrEN · 2021-12-10
> > **Acknowledgment**
> >
> > Thank you to the authors for addressing my few concerns.

---

### Official Review · Reviewer_aLNX · 2021-11-24

**Confidence:** 4
**Overall Score:** 4

**Main Review:**

Advantages:
1)	The two strategies proposed sound reliable.
2)	This paper provides a wealth of cases to illustrate points.
3)	The relevant work provided in this paper is sufficient.

Disadvantages:
1)	Some concepts lack formal definitions, for instance, the "Study Unit".
2)	The strategy proposed in this paper lacks formal expressions and illustrations, which makes the strategy difficult to understand.
3)	Although the strategy proposed in this paper is claimed to be effective in some cases, there is a lack of experiments to prove its effect.
4)	Only two strategies are proposed but the specific implementation is lacking, which makes the contribution may be insufficient
5)	The presentation of this paper is not clear enough and lacks pictures to express opinions intuitively.

**Summary:**

This paper examines the role of textual data as study units when conducting causal inference by drawing parallels between human subjects and organized texts. This paper expounds the main concepts and principles of causal inference, and reveals some vague concepts and fallacies. This paper discusses two strategies in order to facilitate better framing a causal query.

---

> ### Author Response · Authors · 2021-12-03
> **Response to Reviewer aLNX (1/2)**
>
>
> Thanks for reviewing our paper and for your comments. We appreciate the time and effort you put in reviewing, and we found many of your points helpful in revising our paper to be clearer. However, we feel that there might have been a confusion of presentation that led to your comments that (1) the strategies proposed are difficult to understand, and (2) the contributions are insufficient. We have revised our paper to increase its clarity when presenting definitions and introducing those strategies. We hope it addresses your concerns. Please see more details in our point-to-point responses below.
>
>
> > 1. Some concepts lack formal definitions, for instance, the "Study Unit."
>
> Thanks for bringing this to our attention. I n our revision, we will add a formal definition of "study unit": a study unit is an entity subjected to an intervention (often, but not necessarily, independently of other units). We currently give two examples of study units in our article: human subjects (e.g., patients) as often encountered in epidemiology, biomedical sciences, and social sciences, and textual data (e.g., conference papers) as in much recent research at the intersection of machine learning, NLP, and causality. In our revision, we will highlight these two examples of study units.
>
> > 2. The strategy proposed in this paper lacks formal expressions and illustrations, which makes the strategy difficult to understand.
>
> Thank you for raising this question. We have revised our flow when introducing definitions. We would also like to take this opportunity to summarize the two strategies and explain their practical implications, as will be reflected in our revision.
>
> First, instead of imagining manipulating some immutable aspects of textual data, imagine manipulating audiences’ perception of this aspect. This strategy has proven successful in many applications of causal inference for decades. By articulating the timing of the "treatment," researchers can differentiate between "pre-treatment" and "post-treatment" variables: while "pre-treatment" variables need to be adjusted for, "post-treatment" variables should not; in fact, "post-treatment" variables are often "surrogate markers" for the main endpoint or "mediators" which mediate a treatment’s effect.
>
> Second, empirical researchers are advised to shift from one trait to a collection of traits. For instance, if an NLP researcher is interested in how an article’s politeness affects some prespecified outcome, one should treat "readers’ perception of politeness" as a general causal variable encompassing multiple quantifiable traits. The practical implication of this strategy is that, by clearly stating the constituent parts of the intervention, empirical researchers can differentiate between "causal variables" and "pre-treatment variables."
>
> To reliably estimate a causal effect, two strategies described above help frame a causal query, formalize the quantity to be estimated (i.e., the estimand), and assign a role to each variable collected (i.e., causal variable or pre-treatment variable or post-treatment variable). Researchers may then use their favorite estimating strategies such as matching, weighting, outcome modeling via structural equations, and/or a combination of these strategies, to obtain an estimate for the appropriate estimand. In our revised version, we will make these points clearer and discuss the practical implications of the proposed strategies in more detail.
>
> Lastly, we need to mention that both strategies are known to some degree in classical causal inference literature and are state-of-the-art; we are not claiming that we invented them out of the blue. Our main contribution is to identify cases/papers where there are ambiguities/pitfalls framing the causal query and carefully apply these principles to drawing causal conclusions from textual data.

---

> ### Author Response · Authors · 2021-12-03
> **Response to Reviewer aLNX (2/2)**
>
>
> > 3. Although the strategy proposed in this paper is claimed to be effective in some cases, there is a lack of experiments to prove its effect.
>
> Our article is a perspective piece and not a theoretical/empirical paper. However, we do think that the article is a good fit for the conference, as our article aims to clarify "causal principles underlying NLP." While new methods and algorithms are valuable, we feel that clearly stating the foundations/assumptions/estimands is of particular interest for causal inference problems. After all, methods and algorithms are meaningful only when researchers have a sound understanding of what questions these methods and algorithms are going after.
>
> > 4. Only two strategies are proposed but the specific implementation is lacking, which makes the contribution may be insufficient
>
> In this article, we are interested in estimation problems, mostly because this is the primary focus in a lot of recent work at the intersection of NLP and causality; see, e.g., the review paper by Feder et al. (2021) and Keith et al. (2020). Our main contribution in the article is to articulate how to properly frame a causal query, including properly assigning a role to each variable. After figuring out the estimand, the timing of the treatment, causal variables, and pre-treatment variables, researchers may proceed with their favorite estimating strategies, e.g., matching, weighting, outcome modeling via structural equations. Implementation of these estimation strategies is standard and can be found in many off-the-shelf packages/libraries.
>
> > 5. The presentation of this paper is not clear enough and lacks pictures to express opinions intuitively.
>
> We have revised our paper based on your comments and those from other reviewers, and hopefully, the paper is clearer (please see our per-point responses above). Please let us know which part in addition you think is not clear enough or less intuitive. We are happy to clarify them here and revise our paper accordingly (including adding pictures if applicable).
>
>
> ### References
>
> Amir Feder, Katherine A. Keith, Emaad Manzoor, Reid Pryzant, Dhanya Sridhar, Zach Wood-Doughty, Jacob Eisenstein, Justin Grimmer, Roi Reichart, Margaret E. Roberts, Brandon M. Stewart, Victor Veitch, and Diyi Yang. Causal inference in natural language processing: Estimation, prediction, interpretation and beyond, 2021.
>
> Katherine A Keith, David Jensen, and Brendan O’Connor. Text and causal inference: A review of using text to remove confounding from causal estimates. arXiv preprint arXiv:2005.00649, 2020.

---

### Official Review · Reviewer_eiJx · 2021-11-24

**Confidence:** 4
**Overall Score:** 9

**Main Review:**

Let me preface my review by saying that I'm not an expert in NLP nor philosophy of causality (or any kind for the matter). I do however consider myself pretty well versed in machine learning and its intersection with causality, and found this paper very useful.

The paper clarifies many definitions present in the potential outcomes theory and the assumptions that they entail. Furthermore, the authors show how other papers in the ML / NLP literature misuse these terms and implicitly or subtly fail to satisfy the assumptions of the methods they employ. At the very core of it, it shows how people in the machine learning literature constantly misuse the term "causal" when making claims, particularly when no manipulation can be imagined. Of particular importance is the fact that using potential outcomes algorithms without satisfying SUTVA or strong ignorability conditions ( similar to "each unit be potentially exposable to any of the causes") can result in wrong claims, which is seen in machine learning very often. The examples are very easy to understand and it points to very good literature for further clarification.

The paper is very well written, is correct, and extremely precise. It clarifies a lot of misuses and I personally learnt a lot from it. I would love to see a journal version of this paper with more examples of misuse / false claims of causality in the ML literature.

**Summary:**

The paper discusses common misusage of causal tools and language in the machine learning literature, clarifying certain concepts and pointing out important flaws.

---

> ### Author Response · Authors · 2021-12-03
> **Response to Reviewer eiJX**
>
> Thanks for reviewing our paper and for your comments. We appreciate the thoughtfulness of your comments and are glad that you find the article useful in clarifying causal principles. We are currently preparing a manuscript with a detailed case study on drawing causal inference using textual data, and we hope that we may further clarify important causal principles in the context of a detailed example.

---

### Decision · Program_Chairs · 2022-01-12

**Decision:**

Accept (Poster)

**Comment:**

This paper wholeheartedly adopts the potential-outcomes approach to causality and causal inference,  the Holland (1986) slogan of "no causation without manipulation", and the common notion within that school that it is nonsensical to talk about the causal effects of race or sex because those could not imaginably be manipulated while still being the "same person", and then extends these ideas by analogy to text.  On this basis, a lot of things people try to do by way of estimating causal effects of textual features (or using some aspects of texts to control for others, etc.) are indeed nonsensical.  One can also, by analogy, adopt ways people have come up with in the social-scientific literature to evade the Holland/Rubin strictures against ascribing causal effects to race or sex: one can manipulation _perceptions_ of the immutable variables, one can articulate the immutable variables into bundles of usually-associated variables and find aspects which are susceptible to manipulation, etc.

The reviewers found this mostly persuasive, and I suppose I do too, if I can bring myself to accept the original claims about race and sex. I myself find those claims unpersuasive [*], but I recognize that many people in the profession disagree with me.  The analogy on which this paper rests is convincing and well-argued, and many of the suggestions about, e.g., manipulating perceptions are good ideas _even if_ one thinks textual effects are better-defined than they do.

[*]:  These strictures rest on pronouncements by statisticians about what changes would leave a unit "the same person" and what would amount to being "a different person".  These are hard metaphysical questions, far outside our expertise, which we usually avoid, but here they are crucial, and seem to rest merely on intuition about what attributes are part of someone's essence.  (If I were cynical, or a sociologist of science, I'd observe that Holland made his career at the Educational Testing Service, that Rubin consulted for ETS, etc., and that the ETS had a real _interest_ in being able to assert that, scientifically speaking, no one gets a bad score on the SAT _because_ of their race or sex.)  Certainly there are plenty of philosophers of causality, including those who work closely on statistical issues of causal inference like Clark Glymour, who have absolutely no problem with assigning causal effects to race and sex.  (Two citations I happen to have handy: C. Glymour & M. Glymour _Epidemiology_ **25** (2014): 488-490 [https://doi.org/10.1097/EDE.0000000000000122], and  A. Marcellesi, _Philosophy of Science_ **80** (2013): 650--659 [https://doi.org/10.1086/673721].)  Whether  these philosophers' arguments could be extended, analogically, to text is an interesting question that would call for a paper of its own.